# Second Victims in Industries beyond Healthcare: A Scoping Review

**DOI:** 10.3390/healthcare12181835

**Published:** 2024-09-13

**Authors:** Andrea Conti, Alicia Sánchez-García, Daniele Ceriotti, Marta De Vito, Marco Farsoni, Bruno Tamburini, Sophia Russotto, Reinhard Strametz, Kris Vanhaecht, Deborah Seys, José Joaquín Mira, Massimiliano Panella

**Affiliations:** 1Department of Translational Medicine, Università del Piemonte Orientale, 28100 Novara, Italy; 10036607@studenti.uniupo.it (D.C.); 10030824@studenti.uniupo.it (M.D.V.); 20002321@studenti.uniupo.it (M.F.); 20042447@studenti.uniupo.it (B.T.); russottosophia@gmail.com (S.R.); massimiliano.panella@med.uniupo.it (M.P.); 2Doctoral Program in Food, Health, and Longevity, Università del Piemonte Orientale, 28100 Novara, Italy; 3Department of Health Psychology, Miguel Hernandez University, 03202 Elche, Spain; alicia.sanchezg@umh.es (A.S.-G.); jose.mira@umh.es (J.J.M.); 4Wiesabden Institute for Healthcare Economics and Patient Safety, RheinMain UAS, 65197 Wiesbaden, Germany; reinhard.strametz@hs-rm.de; 5Leuven Institute for Healthcare Policy, KU Leuven, 3000 Leuven, Belgium; kris.vanhaecht@kuleuven.be (K.V.); deborah.seys@kuleuven.be (D.S.); 6Atenea Research, FISABIO, 03013 Hermanos López de Osaba, Alicante, Spain

**Keywords:** safety, error management, just culture, peer support, cross-industry learning, second victim phenomenon

## Abstract

The second victim phenomenon (SVP) refers to workers negatively impacted by involvement in unanticipated adverse events or errors. While this phenomenon has been extensively studied in healthcare since its acknowledgment over 20 years ago, its presence and management in other high-risk industries have remained unclear. We conducted a scoping review aiming to map the SVP in non-healthcare industries, as well as to explore the available interventions or support programs addressed to help second victims (SVs). A total of 5818 unique records were identified and, after the screening process, 18 studies from eight sectors were included. All industries acknowledged the existence of the SVP, though many did not use a specific term for defining the SV. Similarities in psychological and emotional consequences were found across sectors. Support strategies varied, with the aviation sector implementing the most comprehensive programs. Self-care and peer support were the most reported interventions, while structured clinical support was not mentioned in any industry. Our review highlighted a lack of standardized terminology and industry-specific, evidence-based support interventions for the SVP outside of healthcare. Healthcare appears to be at the forefront of formally recognizing and addressing the SVP, despite traditionally learning from other high-reliability industries in safety practices. This presents opportunities for reciprocal learning and knowledge transfer between healthcare and other high-risk sectors.

## 1. Introduction

In several work environments, risk factors pose inevitable challenges that directly impact the safety and well-being of people. In response, organizations aim to proactively address these risks through measures that not only physically and emotionally protect their staff but also promote resilience in adverse situations and protect the company’s economy and reputation [1].

These challenges become even more critical in sectors where errors can have serious consequences for users or customers. A clear example of this reality is found in the healthcare field, where patient safety represents a fundamental aspect [2]. As highlighted by the World Health Organization’s “Global Patient Safety Action Plan” [3], it has been estimated that about 10% of the patients are subjected to adverse events while receiving medical care. Moreover, according to the seminal report “To Err is Human: Building a Safer Health System” by the Institute of Medicine [4], a significant proportion of such incidents can be attributed to human errors, highlighting the need to focus on healthcare workers to reduce errors and enhance patient safety. In this context, addressing this problem becomes doubly important since it aims to avoid harm to the patient and also protect the staff involved in human error, who can be significantly affected [2].

As the focus on patient safety and risk factors in healthcare has evolved, it was recognized that healthcare professionals often experience shame and guilt, emotional distress, post-traumatic stress, insomnia, nightmares, lack of self-confidence, burnout, and depression as a consequence of their involvement in clinical errors, thus becoming “second victims” (SVs) [5,6,7]. This term has been coined in the healthcare sector to specifically identify workers negatively impacted by the direct or indirect involvement in unanticipated adverse patient events, unintentional healthcare error, or patient injury [8].

Although the SV phenomenon has been traditionally investigated in the healthcare sector [9], similar occurrences can also be observed in other industries. For example, it has been observed that emotional changes occurring after an incident can tackle airline pilots’ situational awareness and performance [10]. Similar events have been described in oil extraction [11], railway [12], emergency services [13], and teaching [14]. Moreover, the establishment of supportive programs for workers involved in incidents has been described in the literature for both healthcare [5] and other sectors [15,16,17]. On this basis, the conceptual model proposed by Seys et al. outlines a five-stage framework for supporting SVs in which each stage represents a different level of depth and structure in support programs [18].

Despite the similarities across contexts on this phenomenon, theoretical and practical development in different sectors differs significantly. The SV concept has been developed within healthcare, and it has only occasionally been adopted by other industries. On the other hand, other safety-oriented sectors (e.g., aviation, construction) might have made further progress in other areas related to safety and error consequences management. Therefore, exploring this phenomenon outside healthcare might help to understand the SV phenomenon across diverse industries, providing a comprehensive synthesis of available interventions and identifying gaps in the current literature.

This scoping review aims to map and synthesize available knowledge on the SV phenomenon in other industries beyond hospitals or primary care environments, as well as explore the available interventions or support programs addressed to help SVs in these non-healthcare industries.

## 2. Materials and Methods

The present scoping review was designed and conducted following the Joanna Briggs Institute guidelines [19] and reported using the PRISMA-ScR checklist (available as Appendix A) [20]. The PCC (Population, Concept, Context) framework was adopted to construct objectives and selection criteria. The review protocol is available on the Open Science Framework (https://doi.org/10.17605/OSF.IO/8BNHX, accessed on 31 July 2024).

### 2.1. Eligibility Criteria

#### 2.1.1. Population

The population of interest is all active workers, defined as subjects of any sex and age performing activities to produce goods or to provide services for others or themselves [21]. All types of workers were considered, regardless of their role, work duties, or compensation scheme.

#### 2.1.2. Concept

The concept of interest was the acknowledgment of the second victim phenomenon (SVP) in industries different from hospital and primary care facilities. To achieve this objective, we used an operational adaptation of the definition provided by Vanhaecht et al. [8]. Therefore, we considered an SV “any worker, directly or indirectly involved in an unanticipated adverse event (highly stressful event), unintentional error, or customer or user injury, and who becomes victimized in the sense that they are also negatively impacted”.

#### 2.1.3. Context

Errors in the workplace that can cause unexpected and significant harm to the recipients of provided services are not exclusive to the healthcare sector, although the term “SV” has been coined in this context. The opportunities to learn from the experiences of other environments and industries in addressing these situations and promoting worker resilience are the focus of this study. Examples of industries where we initially assumed certain similarities could be found, and therefore opportunities for learning, include the transportation, energy, emergency response, security, and defense sectors, as well as others where the visibility of the impact of an error can have media repercussions, such as Formula (1). Upon reviewing the initial results, we included healthcare personnel working in community pharmacies and instructors and healthcare students in simulations. The first was included because these workers face different challenges compared to in hospital-based pharmacy work. The second was included because the described event was not directly related to the clinical content of the simulation but rather to the pedagogical and procedural aspects, which align more closely with training contexts and the role of the instructor than with traditional healthcare settings and the clinical professional. In this study, we excluded research on human healthcare professionals working in hospitals and primary care from both the public and private sectors.

### 2.2. Types of Sources of Evidence

Due to the exploratory aim of this review, both scientific and gray literature were considered, without restrictions on study design. Only records for which it was not possible to clearly identify the authors were excluded.

### 2.3. Research Questions

What is the current knowledge and understanding of the SVP in non-healthcare industries such as hospital and primary care? What interventions or support programs are available for SVs in these non-healthcare industries?

### 2.4. Search Strategy

The search strategy aimed to gather any relevant report. Google Scholar, Scopus, PubMed, CINAHL, Web of Science, and ProQuest databases were consulted. Search strings are shown in Table 1.

Given the exploratory aim of this scoping review, no publication date or language limits were set so that all relevant records were included. Documents in a language spoken by a member of the research team (i.e., English, Italian, German, and Spanish) were directly assessed, while reports in other languages were translated to English using Google Translator software (Google Inc., Mountain View, CA, USA).

### 2.5. Study Selection and Data Extraction

All the identified records were collected on a Google Sheet database. The screening was performed independently by two reviewers, where inconsistencies were resolved by a third one. Therefore, data extracted from the included report were collected on a standardized form by a researcher (A.C.) and double-checked by another researcher (A.S-G.). The form included the following fields: study design (if any), industry, description of the reported SVP, relevant key findings, and preventive/support program. Included articles were also classified according to the five-level SV support model proposed by Seys et al. [18]. In detail, level 1 represents SVP prevention initiatives for both individuals and organizations; level 2 provides self-care of individual and/or teams; level 3, 4, and 5 structured support programs led by peers and non-clinical and clinical professionals, respectively. Moreover, we added a “level 0” for identifying reports in which the SVP is explicitly acknowledged. The adapted model is shown in Figure 1.

## 3. Results

A total of 11,859 records were retrieved. The complete screening process is shown in Figure 2. Of the fifty screened full texts, 18 studies were included in our review. Among them, four were cross-sectional surveys [22,23,24,25], three case reports [26,27,28], three book chapters [29,30,31], and two qualitative studies [32,33]. Seven studies were conducted in Europe, three in Oceania, two in America, and one worldwide. For five studies, it was not possible to identify the country.

Detailed information about included studies, including the number of citations reported on Google Scholar at the date of the search, is available in Table 2.

While all the identified industries acknowledged the SVP (level 0) and mostly recognized the importance of support by peers and triage (level 3), structured professional support (level 4) was recognized only in aviation. Moreover, no industry described structured clinical support (level 5).

Table 3 shows the number of articles for each industry, mentioning the specific levels of the model shown in Figure 1.

Below, the results are summarized according to their reference sector. Key concepts from the included industries are summarized in Table 4.

### 3.1. Animal Care/Veterinary

Five studies dealt with the animal care/veterinary industry.

The case report by Mosedale et al. [27] described a near-miss event (i.e., wrong patient undergoing a medical procedure) that occurred during a surgical castration procedure performed on a horse. While the authors did not explicitly discuss structured support programs, they mentioned a post-incident team discussion in which the event was debated by all the involved professionals. Moreover, supportive and just culture were mentioned as approaches that can reduce errors, as well as turn incidents into positive learning experiences. While Mosedale et al. did not explicitly refer to the involved veterinary professionals as SVs, they recognize the similarity of the described case to similar ones in a human healthcare context.

A cross-sectional survey conducted in Canada by Perret et al. [25] aimed to assess mental health (e.g., stress, burnout, and anxiety) among veterinarians. As the major results, Perret et al. found high levels of stress, burnout, depression, anxiety, compassion and suicidal ideation. Moreover, they found female professionals had poorer mental health conditions than males. While Perret et al. stated that research on the impact of veterinarians’ mental health on productivity and quality of care is scarce, they recognized a similar phenomenon (i.e, the SVP) in human healthcare. The manuscript did not specifically describe any structured support program; however, the respondents described the support received by family and colleagues as “satisfactory” and “just satisfactory”, respectively.

The qualitative study performed by Gibson et al. [32] aimed to provide insights into veterinarians’ experiences related to adverse events, specifically regarding ethical challenges. Participants reported that, in the aftermath of an adverse event, they experienced concern about taking responsibility, as well as difficulties in balancing honesty, apology, and compensation toward clients. Moreover, Gibson et al. suggested that such evidence would foster the future development of veterinarian support programs aimed at reducing emotional impact associated with such challenges and improving patient safety. In addition, it was explicitly stated that ethical challenges and moral distress can be associated with the development of SV in veterinary practice. Moreover, the article stated the potential benefits of group discussions as a strategy to reduce the SVP, despite the fact that no practical examples of such an approach were given.

Timmenga et al. [38] investigated the design and impact of support programs for veterinarians toward mental well-being. This mixed method study included a quantitative online survey, interviews with two focus groups, and input gathered after webinars. They identified that many animal care organizations established support programs both at individual and workplace levels. In addition, support from colleagues/relatives after an adverse event was reported as a coping strategy. In contrast, no structured peer program was reported.

White et al. [39] explored both contributing factors and interventions that aimed to mitigate risk and increase personal well-being among veterinarians. They observed that the literature on structured group/peer support for veterinarians is limited. Moreover, they stated that the stressors, coping strategies, and roles of the animal care sector are comparable to the human one, from which veterinarians can translate knowledge on personal well-being. Therefore, White et al. developed a specific training course aimed at enhancing veterinary staff knowledge of burnout and fatigue. Interestingly, they declared an additional professional experience in different critical industries, namely wildlife care and oil spill response events. While the article supported the introduction of such peer program, it also emphasized the logistic barriers that could hamper participation, for example, the fact that most veterinarians are freelance or work in small clinics. Although the described scenarios were compatible with the SVP definition, White et al. did not use the SV term in their manuscript.

### 3.2. Aviation

Two studies were conducted in the aviation sector.

The cross-sectional study conducted by Karanikas et al. [24] used a questionnaire to explore whether just culture policies could be introduced to a military aviation organization operating in Europe. Moreover, they introduced the SV concept within the studied sector, adopting the definition of Dekker [40]. Respondents were propositive toward psychological support and short-term leave after an incident. However, the authors concluded that the implementation of a just culture policy, within this specific organization, could be not directly feasible.

Apanay [34] explicitly translated the SV concept from healthcare to aviation. The study aimed to discuss the SVP in the aviation industry and to identify possible strategies to prevent SVs. A parallelism between healthcare and other safety critical industries was made: both contexts, indeed, have been described as complex organizations in which safety is of paramount importance. With regards to prevention, Apanay identified in crew resource management a potential tool to support SV. Moreover, he advocated early access to supportive interventions aimed at supporting SV. Behavioral, psychological, and cultural/organizational barriers (namely, strategies aimed at mitigating the phenomenon) were identified to prevent the progression of an SV to a more severe condition. Behavioral barriers were identified in a strong social group-based support (e.g., family, friends, colleagues/managers). The psychological barriers were identified in tools, such as the “critical stress incident management” approach, used to mitigate the acute psychological distress. In addition, a structured peer support system of colleagues has been identified. Finally, organizational culture barriers were identified in just culture or positive safety culture.

### 3.3. Business

Only one study dealt with business.

The chapter “Evaluating Just Culture and Restorative Practices: The Business Case” from the book *“Restorative Just Culture in Practice”* [41] discussed the SVP within a description of restorative just culture. In detail, they identified SV as the target for empowerment within a just culture approach. However, although the chapter identified the existence of the SVP within the business sector, it did not provide any defined strategy for dealing with it.

### 3.4. Construction

Two studies were conducted in the construction sector.

Heraghty et al. [23] cited the SV theory by analyzing the writing styles of an incident report to assess whether it can influence the proposed different solutions. In detail, 93 construction workers were asked to provide preventive/corrective recommendations after reading an accident report. They stated that the report style can have a significant impact on the outcome of the analysis, and therefore, a just culture is needed to maintain a positive learning culture.

The same research team [26] published an action research study to assess the approach to accident learning and justice mechanisms. An accident analysis process was introduced in two construction companies, and workers were asked to fill a survey to express their opinions about the newly introduced procedure. They found that a restorative, focused process can benefit both the workers and the organization. In detail, the absence of any punishment supported an open disclosure of accidents. The authors explicitly stated that this approach minimized the SVP, with positive benefits.

### 3.5. Engineering

Three studies dealt with the engineering industry.

Hayes et al. [22] explored decision-making in this sector and, more specifically, the defensive engineering through a questionnaire proposed to Australian professionals. They observed that, if a defensive approach in the human healthcare sector was a widely studied topic, the research in defensive engineering was scarce. They observed that engineers adopt defensive engineering strategies to protect themselves from disaster consequences; however, such measures were not always aligned with the best strategies for disaster prevention. While discussing moral and legal responsibilities, Hayes et al. argued that not only engineers involved in legal issues were impacted by the negative consequences of the event but also other professionals closely observing the event. They concluded that engineers were developing problems and issues similar to the SVP in healthcare, despite the fact that the phenomenon in this sector was not already well defined.

The book published by Smith et al. [31] discussed cognitive systems engineering, and an entire chapter (“Speaking for the Second Victim”) is dedicated to SVs. They explicitly adopted the definition made by Dekker [40]. The chapter discussed the approach to understanding SVs’ point of view, when the SV can not express themself (i.e., he/she is deceased). While Smith et al. did not provide any additional information about how to prevent or manage SVs, they underlined the importance of listening to the SV’s experience.

The qualitative study by van Marrewijk et al. [35] aimed to investigate organizational processes following a critical incident. The authors recognize the existence of the SVP in the construction sector. They commented that the investigation of the incidents acted as a “humiliating and anxious experience” for SVs. Moreover, they identified the role of the organization in protecting and supporting the SVs after an event.

### 3.6. Pharmacy

Only one study was retrieved from the pharmacy sector.

The editorial by Phipps et al. [36] proposed strategies and approaches that aimed to manage dispensing-error risk in community pharmacies. They explicitly mentioned the SVP, adopting the definition by Scott et al. [42]. Moreover, the article highlighted the importance that SVs are treated using a supportive approach, and they mentioned the components of a potential support program for SVs (e.g., safe space for reviewing the accident and basic peer and professional support).

### 3.7. Simulation (Training)

One study dealt with simulation.

The case report by Naweed et al. [28] dealt with SVs in simulation. The authors highlight how instructors and facilitators were exposed to adverse events during their activity in a manner similar to the clinical practice. The report described a panic attack of a facilitator during a simulated evacuation scenario. Therefore, how facilitators “are equally vulnerable to breaches in simulation safety (physical or psychological) as their learners” were highlighted. In conclusion, they stated the importance of self-care and open error disclosure for simulators as well.

### 3.8. Social Workers and Child Protection Systems

Three studies dealt with social workers and child protection systems (CPS).

The mixed method study by Haight et al. [37] discussed the moral injury phenomenon among child protection professionals. They collected qualitative and quantitative information through surveys among the professionals working in CPS of two different counties to assess whether professionals can experience moral injury regarding their work within CPS. Some parallelisms with other sectors were made and they identified SVs as healthcare professionals experiencing a moral injury. Study participants described morally injurious events as “occurring when actions or inactions by people or systems entrusted with helping or protecting others instead caused harm”. The majority of them identified a lack of resources in their organization as a major cause of the occurrence of morally injurious events. Moreover, they identified issues in laws, policies, and procedures. With regard to the professionals’ psychological response, they reported emotional distress and existential issues. They identified quitting the job as a major consequence of moral injury.

The qualitative study by Kvitsiani et al. [33] aimed to explore the dynamic of moral injury among social workers. Similar to Haight et al. [37], they identified a parallelism between moral injury and SVP. They conducted a semi-structured interview and explored the situations causing a moral injury, emotional reactions, and personal changes.

The book *“Stress, Trauma, and Decision-Making for Social Workers”* by Regehr [30] explicitly introduced the concept of SV. In detail, in the chapter “Factors contributing to trauma response”, they mentioned that ongoing stressors related to an incident can exacerbate personnel traumatic reactions in firefighters and paramedics, as well as police officers. Next, they identified an SVP similar to the experiences of social workers, even if they did not adopt SV terminology for the social context. As possible mitigation interventions, they identified peer support teams as a strategy that, similarly to the SVP, can support morally injured professionals.

## 4. Discussion

In healthcare, the SV phenomenon was acknowledged more than 20 years ago [43]. Since then, several studies have been conducted to better understand and define the phenomenon, as well as support programs have been conceptualized and developed [44]. To date, the SV term is currently widely used in the literature to identify this phenomenon among healthcare professionals [5,45,46].

Our review aimed to identify and characterize the SVP, as well as its support strategies, in other sectors. While our search was not limited to a specific sector, only eight industries were represented among our results.

All of the industries recognized the existence of the SVP, despite several of them not adopting the SV term. Moreover, it was not possible to identify an alternative, standardized, and widely acknowledged term for describing this phenomenon. This absence of a standardized and shared conceptualization and vocabulary might hamper the translation of knowledge and support strategies across different industries. However, our findings reveal similarities in the psychological and emotional consequences experienced by SVs across diverse sectors, which have been studied particularly in the healthcare context [47]. For example, workers from the aviation, engineering, veterinary, and social sectors reported symptoms such as distress, anxiety and self-doubt, in line with findings described in healthcare [48]. Such similarities suggest that the SVP might transcend the boundaries of any specific industry, work context, or human condition.

With regards to support programs for SVs, our review identified and classified evidence from several sectors. In detail, four out of the five support levels were identified in the aviation industry. This is in line with the long-lasting culture of a non-punitive environment in this sector, aimed not only to increase safety [49] but also to improve crew performance [50]. The most reported support strategies were self-care and peer support, while professional clinical support was not mentioned in any industry. Moreover, such experiences were anecdotal and described only in general terms. However, several industries successfully implemented programs that aimed to mitigate the consequences of critical incidents on staff. For example, the airline industry developed the “critical incident stress management” (CISM) program, which—similarly to SV support from healthcare—aimed to help workers cope with the psychological impact of incidents. Although such programs have been developed over the past 50 years [51], only in 2020 did they become mandatory for all air traffic and navigation services in the European Union [52].

Despite identifying awareness and support programs in industries such as oil/gas production, air traffic control, detainment, ground military services, and railway, our review did not find specific studies from these sectors. Three potential reasons could explain this gap: a lack of conceptualization of root causes, an intentional effort to keep such data undisclosed, or the use of different taxonomies focusing on concepts like moral distress or moral injury. Reasons two and three are the most likely. The lack of industry-specific, evidence-based support interventions tailored to the unique needs of SVs in different work contexts represents a significant gap that needs to be addressed through further research and program development. Several potential barriers and challenges in addressing the SV phenomenon across non-healthcare industries can be inferred from the findings. Firstly, the absence of a standardized definition and terminology can impede efforts to raise awareness and prioritize this issue within different organizations and sectors. Secondly, the limited availability of industry-specific support programs and interventions tailored to the unique needs of SVs in diverse work contexts poses a challenge in effectively mitigating the negative consequences. Additionally, organizational and cultural factors, such as the reluctance to acknowledge errors or the presence of punitive environments, can create barriers to open communication and seeking support [53]. Furthermore, logistical and resource constraints, particularly in industries with decentralized or remote operations (e.g., veterinary practices, and engineering projects), may hinder the implementation and accessibility of support programs for SVs [54].

Interestingly, while the healthcare sector has historically learned from other high-reliability industries like aviation in areas such as crew resource management and safety culture [55], the findings of this review indicate that healthcare may be at the forefront when it comes to formally recognizing and addressing the SVP. The concept of the SV originated in healthcare [43], along with clear definitions and theoretical models for understanding and supporting these affected workers. In contrast, non-healthcare sectors appear to lack such standardized terminology, formal definitions, and comprehensive support frameworks specifically tailored to their SVs. This paradox highlights how healthcare has taken a pioneering role in explicitly acknowledging the SV experience and establishing a knowledge base to develop appropriate interventions and support systems. Therefore, while healthcare can continue to adopt best practices from other industries in areas like safety management [55], the sector’s advancements in the SV domain may offer valuable lessons and opportunities for knowledge transfer to other high-risk, non-healthcare fields.

However, our scoping review presents some limitations. First, the lack of standardized definitions for the SVP across different industries could have led to missing some studies that used different terms or conceptualizations to describe similar experiences. Additionally, the small number of included studies and their heterogeneity in terms of study design, industry focus, and depth of analysis might limit the generalizability of the findings. Third, although we consulted databases including grey literature (e.g., Google Scholar), our review relied on published literature and may not capture unpublished or internal industry practices addressing the SVP.

## 5. Conclusions

This scoping review explored the SVP in non-healthcare industries. Various sectors, including aviation, engineering, and veterinary services, recognize the psychological impacts of adverse events on workers, often adopting different terminologies. These industries have implemented support strategies like CISM in aviation and peer support systems in veterinary practice, offering valuable insights for the healthcare industry. Healthcare, traditionally learning from high-reliability industries, can adopt these support strategies. Aviation’s CISM programs and engineering’s focus on moral distress provide frameworks that healthcare can adopt. The veterinary sector’s emphasis on ethical challenges also offers lessons for healthcare’s SV support. In all the included industries, support strategies were identified, particularly regarding individual/team self-care and peer support. Despite healthcare’s progress in recognizing the SVP, reciprocal learning from other industries can enhance support programs. Future research should focus on integrating these industry-specific interventions into healthcare, addressing barriers like organizational resistance and logistical constraints. Cross-industry collaboration and knowledge sharing can improve support for SVs, enhancing safety and well-being across all sectors.

## Figures and Tables

**Figure 1 healthcare-12-01835-f001:**
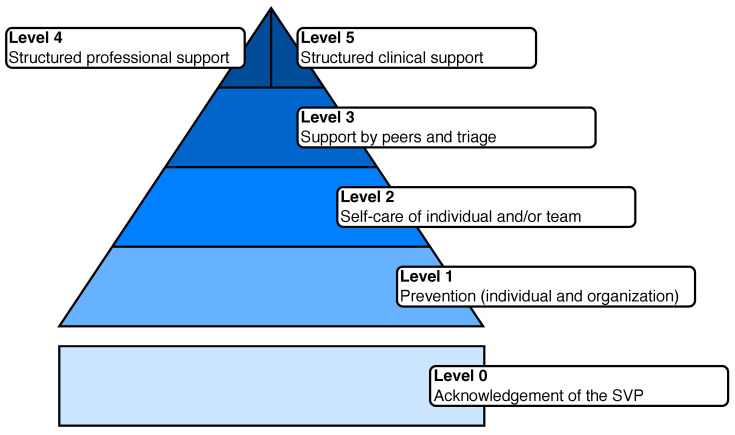
The modified version of the five-step model (adapted from Seys et al. [18]).

**Figure 2 healthcare-12-01835-f002:**
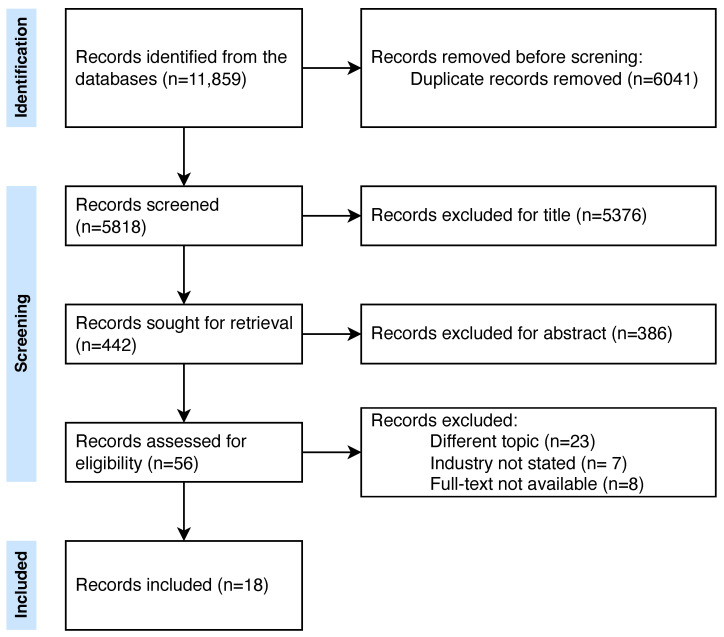
PRISMA flowchart.

**Table 1 healthcare-12-01835-t001:** Search strategy.

Database	String	Date	Results
Google Scholar	allintitle: “second victim” OR “second victims”	5 June 2024	470
Google Scholar	“second victims”	5 June 2024	7571
ProQuest	title (“second victim” OR “second victims”) OR abstract (“second victim” OR “second victims”)	5 June 2024	162
PubMed	“second victims” OR “second victim”	5 June 2024	476
Scopus	“second victims” OR “second victim”	5 June 2024	2287
CINAHL	“second victims” OR "second victim”	5 June 2024	317
Web of Science	“second victim” OR “second victims” (Title) OR “second victim” OR “second victims” (Abstract)”	5 June 2024	593

**Table 2 healthcare-12-01835-t002:** Included studies.

First Author	Year	Study Design	Country	Sector	Levels	Citations *
Apanay [34]	2021	Opinion paper	-	Aviation	0, 1, 2, 3, 4	0
Karinakas [24]	2017	Survey	Europe	Aviation	0, 4	7
De Boer [29]	2022	Book	-	Business	0	0
Heraghty [23]	2018	Survey	Australia	Construction	1	12
Heraghty [26]	2021	Case study	United Kingdom	Construction	1	10
van Marrewijk [35]	2024	Qualitative	Holland	Construction	0, 2	1
Hayes [22]	2018	Survey	Australia	Engineering	0	5
Smith [31]	2017	Book	-	Engineering	0	29
Phipps [36]	2020	Editorial	-	Pharmacy	0, 2, 3	1
Naweed [28]	2021	Case report	Austria	Simulation	0	6
Haight [37]	2017	Mixed-method	United States	Social-CPS	0	112
Kvitsiani [33]	2023	Qualitative	Georgia	Social-CPS	0	0
Regehr [30]	2018	Book	-	Social-CPS	0, 3	41
Gibson [32]	2003	Qualitative	United Kingdom	Veterinary	0, 2	0
Mosedale [27]	2020	Case report	United Kingdom	Veterinary	0, 2	1
Perret [25]	2020	Cross-sectional	Canada	Veterinary	0, 2	84
Timmenga [38]	2022	Mixed-method	Worldwide	Veterinary	2, 3	8
White [39]	2021	Pilot	New Zealand	Veterinary	2, 3	13

* The number of citations was gathered from Google Scholar.

**Table 3 healthcare-12-01835-t003:** Intersection between industries and levels (number of articles mentioning the specific level/total articles). Level 0: acknowledgment of the SVP; Level 1: prevention (individual and organization); Level 2: self-care of individual and/or team; Level 3: support from peers and triage; Level 4: structured professional support; Level 5: structured clinical support.

Industry	Level 0	Level 1	Level 2	Level 3	Level 4	Level 5
Veterinary	3/5	0	5/5	2/5	0	0
Aviation	2/2	1/2	1/2	1/2	2/2	0
Social-CPS	3/3	0	0	1/3	0	0
Engineering	3/3	0	0	1/3	0	0
Construction	1/2	2/2	1/2	0	0	0
Business	1/1	0	0	0	0	0
Pharmacy	1/1	0	1/1	1/1	0	0
Simulation	1/1	0	0	0	0	0
Overall	13/18	3/18	7/18	5/18	2/18	0/18

**Table 4 healthcare-12-01835-t004:** Emerging key concepts from the included industries.

Industry	Key Concepts
Aviation	Behavioral, psychological, and organizational barriers are crucial in preventing SV progression in aviation.
	Just culture implementation in military aviation is challenging and may not be directly feasible.
Business	The existence of the SVP in the business sector is acknowledged, but there is a lack of defined strategies for addressing it.
Construction	Implementing a restorative-focused accident analysis could mitigate the SVP and promote open disclosure
Engineering	Similarly to human healthcare, engineers adopt defensive strategies that may not align with optimal disaster prevention.
	Critical incident investigations can be humiliating for SVs, which should be protected and supported by organizations.
Pharmacy	The SVP was acknowledged in this industry and was linked to the dispensing errors.
Simulation	Both instructors and facilitators can be subjected to adverse events, similar to clinical practice.
Social-CPS	Child protection and social work professionals can experience moral injuries similar to SVP.
	Peer support is a potential strategy to mitigate moral injuries.
Veterinary	Veterinarians face significant stress, burnout, anxiety, and ethical challenges related to adverse events, similar to human healthcare.
	While peer and colleague support exists, there are few structured support programs for veterinarians.

## Data Availability

Data are contained within the article.

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
