# Peer review of "Second Victims in Industries beyond Healthcare: A Scoping Review"

_healthcare, 2024, doi:10.3390/healthcare12181835_

Round 1
Reviewer 1 Report
Comments and Suggestions for Authors
This scoping review from Conti et al. aims to map and synthesize available knowledge on the second victim phenomenon (SVP) in other sectors/industries beyond hospitals or primary care environments. Moreover, the authors explored the available interventions or support programs addressed to help SVs in these non-healthcare sectors.
The present scoping review was designed and conducted following the Joanna Briggs Institute guidelines. The scoping review was reported using the PRISMA-ScR checklist. The PCC (Population, Concept, Context) framework was adopted to construct objectives and selection criteria. Methodologically, the authors have proceeded very well. The authors considered both academic and gray literature in this review. The gray literature in particular can contain a lot of valuable information about intervention programs and company health promotion. No publication date or language limits were set.
The research questions are well derived from the literature and formulated precisely.
The authors answer the questions (“What is the current knowledge and understanding of the SVP in non-healthcare industries such as hospital and primary care? What interventions or support programs are available for SVs in these non-healthcare industries?”) completely.
A few inconsistencies in the table and the text should be noted here:
In Results it says: “The majority of the studies were conducted in Europe, while it was not possible to identify the country for seven of them.”
But the paper
[38] Organizational learning from construction fatalities: Balancing juridical, ethical, and operational processes
Alfons van Marrewijk, Hans van der Stehen
Safety Science, Volume 174, June 2024, 106472
https://doi.org/10.1016/j.ssci.2024.106472
(Page 3, under “3. Methods”) with supposedly incomplete information about the country says the following:
“A longitudinal qualitative research approach was used to study safety meetings in the Dutch construction sector and the Gebr. van der Steen case between 2018 and 2023.”
So, it's a Dutch population.
In the next paper
[36] Moral injury among Child Protection Professionals: Implications for the ethical treatment and retention of workers
Wendy Haight, Erin P. Sugrue, Molly Calhoun
Children and Youth Services Review, Volume 82, November 2017, Pages 27-41
https://doi.org/10.1016/j.childyouth.2017.08.030
under 2.2:
“Site Participants in this study were sampled from two adjacent counties in a metropolitan area of a Midwestern state. In this state, the child welfare system is administered by the counties. Willow County1 has a population of approximately 1.2 million people. … Willow County has a large refugee and immigrant population with approximately 13% of residents being foreign-born. … Approximately 46% of the population holds a bachelor's degree or higher (U.S. Census Bureau, 2016). Spruce County has approximately 500,000 residents…. The median household income in Spruce County is $55,000 annually with 17% of the population living in poverty. Approximately 40% of the population hold a bachelor's degree or higher (U.S. Census Bureau, 2016). ….”
So, it's a U.S. population.
Small comments:
Table 1: in the lines PubMed and Scopus the quotation marks are missing in the string „second victim“.
The information on the reference [34] is not complete. The information on the year and DOI is missing.
Apanay…
Examining the Problems of Creating Second Victims in the Aviation Industry
June 2021
DOI: 10.13141/RG.2.2.23484.34364
In summary, this systematic review is very important for the Healthcare.
If the authors can improve the manuscript by minor revisions, it is recommended to publish it in Healthcare.
Author Response
Dear Reviewer 1,
We wish to thank the Reviewer for the effort in assessing our manuscript. We confirmed we have corrected all the minor inconsistencies (highlighted in red in the manuscript) pointed out by him/her. For the reference #34 (Apanay 2021), we inserted the missing information (month/year and DOI) in the BibTeX file, and we will ask MDPI editorial staff to fix also the PDF.
Reviewer 2 Report
Comments and Suggestions for Authors
This research about 'second victims' in industries other than healthcare provides advice for industries to consider SV as a preferential area. Despite heterogeneity in studies, the authors have discussed the SV with the consolidation. The authors followed the PRISMA thoroughly to prepare the scoping review.
I have gone through all sections and verified the required information to be included (PRISMA as well as otherwise).
I would highly recommend accepting the scoping review in its present form.
Additional comments:
- The mandate of a scoping review is to find the gaps that need to be filled to create a more plausible working environment within an industry. The present study picked an exciting and important pain point for the industries to have a clear phenomenon/regulation for a second victim. It identifies the industries not having such a phenomenon. I can understand that due to its absence, an optimum level of satisfaction is never achievable.
- Most studies in the healthcare field were found to use the keyword 'second victim'. Other industries were found very rarely.
- In the manuscript, I found that PRISMA has been followed perfectly.
- the conclusion is exclusively in line with the main question.
- Tables and figures are very easy to understand.
Author Response
Dear Reviewer 2,
We would like to thank the reviewer for his/her efforts in reviewing our manuscript. According to his comment, we improved the conclusion (changes are in red).
Reviewer 3 Report
Comments and Suggestions for Authors
First of all, I would like to say that I enjoyed reading the study, that I think it addresses a very relevant concept, and that the development of the research brings new insights and highlights (and shadows) its potential and usefulness as a tool in different fields. I will try to justify this general assessment with more specific aspects, complementing it with a series of suggestions (not criticisms) to improve this remarkable study.
Introduction. It perfectly contextualises the study, placing it fundamentally in the field of health, although I would have liked to see more details, for example on the genesis of the study or other reference studies. As for the aim, I would try to formulate it in a more direct way (trying to create a more integrated paragraph, lines 55 to 62)
Materials and methods. I understand that the authors say that the research team directly reviewed publications in 4 languajes. This is a value that is stated in the negative (language not spoken, line 112 and 113), I would rephrase it in the positive. On the other hand, regarding the content of the question, I think that the search strategy (Table 1) and the
Results. The search strategy (Table 1) and the results (Table 3) are very clear and informative. Personally, I think that Table 3 is sorted alphabetically by author, which is not very informative. It would be much more relevant to sort it by sector or by year. On the other hand, this section contains a synthesis of the studies, which I understand is limited (in terms of space and words) by the nature of the journal. Nevertheless, it provides the basic keys to the studies found. I believe that this could be strengthened in some way by a table that includes, by sector, the main elements of the review of the studies analysed. I think this would provide an overview for discussion.
Discussion. I think it highlights relevant aspects and puts its finger on the sore spot, for example by showing that high reliability industries (energy, air traffic control...), pioneers of safety culture or human factors programmes, have hardly developed studies on the concept of "second victims". It is, as one might expect, a discussion that is speculative in its ideas but narrow in its analysis.
Finally, I would like to point out that the above should be seen as a suggestion and in no way detracts from the high value of the study in my opinion. Congratulations to the authors!
Author Response
Dear Reviewer3,
Thank you so much for your effort in reviewing our manuscript. Please find our answers below, in red.
Comment: First of all, I would like to say that I enjoyed reading the study, that I think it addresses a very relevant concept, and that the development of the research brings new insights and highlights (and shadows) its potential and usefulness as a tool in different fields. I will try to justify this general assessment with more specific aspects, complementing it with a series of suggestions (not criticisms) to improve this remarkable study.
Introduction. It perfectly contextualises the study, placing it fundamentally in the field of health, although I would have liked to see more details, for example on the genesis of the study or other reference studies. As for the aim, I would try to formulate it in a more direct way (trying to create a more integrated paragraph, lines 55 to 62)
Reply: We revised this paragraph.
Comment: Materials and methods. I understand that the authors say that the research team directly reviewed publications in 4 languajes. This is a value that is stated in the negative (language not spoken, line 112 and 113), I would rephrase it in the positive.
Reply: We have rephrased the sentence.
Comment: On the other hand, regarding the content of the question, I think that the search strategy (Table 1) and the results (Table 3) are very clear and informative. Personally, I think that Table 3 is sorted alphabetically by author, which is not very informative. It would be much more relevant to sort it by sector or by year.
Reply: According to the reviewer's suggestion, we resorted Table 3 by sector.
Comment: On the other hand, this section contains a synthesis of the studies, which I understand is limited (in terms of space and words) by the nature of the journal. Nevertheless, it provides the basic keys to the studies found. I believe that this could be strengthened in some way by a table that includes, by sector, the main elements of the review of the studies analysed. I think this would provide an overview for discussion.
Reply: We agree with the reviewer. We added a table with key information for each sector (the new Table 4).
Comment: Discussion. I think it highlights relevant aspects and puts its finger on the sore spot, for example by showing that high reliability industries (energy, air traffic control...), pioneers of safety culture or human factors programmes, have hardly developed studies on the concept of "second victims". It is, as one might expect, a discussion that is speculative in its ideas but narrow in its analysis.
Finally, I would like to point out that the above should be seen as a suggestion and in no way detracts from the high value of the study in my opinion. Congratulations to the authors!
Changes in the manuscript are highlighted in red.
Reviewer 4 Report
Comments and Suggestions for Authors
A nice review on SVP. I believe it will contribute to the literature.
Have previous reviews been shown in the article?
It would be useful to add the following addresses to the indexes that are scanned.
The Cochrane Library www.cochrane.org
The Joanna Briggs Institute www.joannabriggs.edu.au/pubs/systematic_reviews.php
The Campbell Collaboration www.campbellcollaboration.org
The Centre for Evidence-Based Medicine www.cebm.net
The NHS Centre for Reviews and Dissemination www.york.ac.uk/inst/crd
Bandolier www.medicine.ox.ac.uk/bandolier
Author Response
Dear Reviewer 4,
A nice review on SVP. I believe it will contribute to the literature.
Have previous reviews been shown in the article?
It would be useful to add the following addresses to the indexes that are scanned.
The Cochrane Library www.cochrane.org
The Joanna Briggs Institute www.joannabriggs.edu.au/pubs/systematic_reviews.php
The Campbell Collaboration www.campbellcollaboration.org
The Centre for Evidence-Based Medicine www.cebm.net
The NHS Centre for Reviews and Dissemination www.york.ac.uk/inst/crd
Bandolier www.medicine.ox.ac.uk/bandolier
Reply: We wish to thank the reviewer for the time spent in revising our manuscript. As known by the Authors, no previous review on this topic were published. We performed an explorative search on different databases (i.e., PubMed, Scopus, Google Scholar) to confirm this point. Moreover, we did not screen the indexes suggested by the reviewers as the focus of our review was the non-healthcare industries.